# Computerized Cytological Features for Papillary Thyroid Cancer Diagnosis—Preliminary Report

**DOI:** 10.3390/cancers11111645

**Published:** 2019-10-25

**Authors:** Shyang-Rong Shih, I-Shiow Jan, Kuen-Yuan Chen, Wan-Yu Chuang, Chih-Yuan Wang, Yung-Lien Hsiao, Tien-Chun Chang, Argon Chen

**Affiliations:** 1National Taiwan University College of Medicine, Taipei 10051, Taiwan; srshih@ntu.edu.tw (S.-R.S.); cyw1965@gmail.com (C.-Y.W.); 2Department of Internal Medicine, National Taiwan University Hospital, Taipei 10002, Taiwan; hsaio17@yahoo.com.tw; 3Center of Anti-Aging and Health Consultation, National Taiwan University Hospital, Taipei 10002, Taiwan; 4Department of Laboratory Medicine, National Taiwan University Hospital, Taipei 10002, Taiwan; isjan15719@ntu.edu.tw; 5Department of Surgery, National Taiwan University Hospital, Taipei 10002, Taiwan; dtsurg51@gmail.com; 6AmCad BioMed Corporation, Taipei 10547, Taiwan; rachel.chuang@amcad.com.tw; 7Far Eastern Polyclinic, Taipei 10043, Taiwan; 8Institute of Industrial Engineering, National Taiwan University, Taipei 10617, Taiwan

**Keywords:** thyroid cytology, diagnosis, computer analysis, morphometry

## Abstract

Fine needle aspiration cytology (FNAC) is the final diagnosis of thyroid nodules before surgery. It is important to further improve the indeterminate FNAC diagnosis results using computerized cytological features. This retrospective cross-sectional study included 240 cases, of whom 110 had histologic diagnosis of papillary thyroid cancers (PTC), 100 had nodular/adenomatous goiters/hyperplasia (benign goiters), 10 had follicular/Hurthle cell carcinomas, and 20 had follicular adenomas. Morphological and chromatic features of FNAC were quantified and analyzed. The result showed that six quantified cytological features were found significantly different between patients with a histologic diagnosis of PTC and patients with histologic diagnosis of benign goiters in multivariate analysis. These cytological features were used to estimate the malignancy risk in nodules with indeterminate FNAC results. The Area Under the Receiver Operating Characteristics (AUROC) of the diagnostic accuracy with a benign or malignant nature was 81.3% (*p* < 0.001), 78.7% (*p* = 0.014), and 56.8% (*p* = 0.52) for nodules with FNAC results of atypia, which is suspicious for malignancy and follicular neoplasm, respectively. In conclusion, quantification of cytological features could be used to develop a computer-aided tool for diagnosing PTC in thyroid nodules with indeterminate FNAC results.

## 1. Introduction

Thyroid nodules are common. They can be detected in 19%–67% of the normal population using high-resolution sonography, and 5%–15% are malignant [1]. Fine needle aspiration cytology (FNAC) has an essential role in evaluating thyroid nodules before surgery. In samples satisfactory for interpretation, 2%–5% are reported as definitively malignant and 55%–74% as definitively benign [2,3,4,5]. The remaining samples have indeterminate cytology, including atypia of undetermined significance or follicular lesion of undetermined significance (AUS/FLUS) (2%–18%), follicular neoplasm or suspicious for a follicular neoplasm (FN/SFN) (2%–25%), and suspicious for malignancy (SUSP) (1%–6%) [6]. Therefore, researchers are seeking methods to further improve preoperative diagnosis or prognostic prediction of thyroid nodules, including genetic evaluation and computerized cytological morphometry [6,7,8]. Commercial molecular tests for thyroid FNAC include ThyroSeq, Afirma, RosettaGX Reveal, ThyGenX, and ThyraMIR [9]. Their sensitivity for the diagnosis of thyroid cancer ranges from 74% to 91%, with specificity from 49% to 93%, negative predictive value (NPV) from 92% to 97%, and positive predictive value (PPV) from 37% to 83% in tumors with cytological categories of AUS/FLUS and FN. The molecular approach is an ideal “rule-in” test if the PPV for histopathologically-proven malignancy is high. On the other hand, the molecular approach is an ideal “rule out” test if the NPV is high. Most molecular tests listed above have high NPV and lower PPV [6]. The reproducibility of some genetic tests measured in different populations has been questioned [10,11,12]. Moreover, the molecular tests are expensive and not widely available. Computerized cytomorphometry enables objective quantification of selected morphologic and chromatic parameters in individual cells [13]. It serves as a sequential reader to the conventional cytological evaluation. We aimed to analyze the cytological features of benign and malignant thyroid nodules using computerized cytomorphometry to aid in the differential diagnosis of FNAC. Papillary thyroid cancer (PTC) is the most common thyroid malignancy and constitutes up to 90% of differentiated follicular cell-derived thyroid cancers (FCTC) worldwide [14]. In this preliminary study, we focused on the cytological features of PTC and the other type of FCTC, namely, follicular thyroid cancer (FTC), in comparison with those benign follicular cell-derived lesions.

## 2. Results

Table 1 shows demographic characteristics, according to categories of cytological diagnoses. Malignant tumors accounted for 13.33%, 57.5%, 31.25%, 94.34%, and 100%, respectively, of the benign, AUS, FN/SFN, SUSP, and malignant categories on cytological diagnosis. Patients’ age and sex were not significantly different between the pathologically diagnosed benign and malignant tumors in each of the cytological categories. In patients with pathologic diagnosis of PTC, five cases had follicular variants of PTC (FVPTC). Three of them had non-encapsulated subtype of FVPTC (NFVPTC) and two had mixed classical type of PTC (cPTC) and NFVPTC. The other 105 PTCs were cPTC.

### 2.1. Diagnosis of Papillary Thyroid Cancer (PTC) and Benign Follicular Lesions Other than Adenoma

In pathologic diagnosis of PTC and benign nodular/adenomatous goiter/hyperplasia, eight cytological features in FNAC specimens were significantly different between pathologically benign and malignant nodules, including mean nuclear size (MNS), standard deviation of nuclear size (SDNS), mean of nuclear elongation (MNElon), coefficient of variation of nuclear elongation, nuclear-cytoplasmic saturation ratio (NCSR), nuclear-cytoplasmic ratio (NCR), nuclear polarity (NP), and inclusion Index (II) (Table 2).

These eight features were then analyzed using stepwise logistic regression with the *p*-value of entering variables set at 0.05 and the *p*-value of removing variables set at 0.1. Six features, i.e., MNS, MNElon, NCSR, NCR, NP, and II, remained significant and were retained in the model. A logistic regression model was then built using these six cytological features as independent variates and pathologic diagnosis as the dependent variate (benign: 0, malignant: 1) (Table 3).

With the logistic regression model, the probability of malignancy could then be estimated using the following. Probability = $\frac{1}{1+e^{-log\;it\left(p\right)}}$ where *logit*(*p*) = −9.71158 + 0.06557MNS + 8.94005MNElon − 1.27953NCSR + 1.895NCR + 0.038098NP + 32.1987II.

With the estimated probability of malignancy calculated for each case, the receiver operating characteristic (ROC) curve analysis was performed. In pathologic diagnosis of papillary thyroid cancer and benign follicular lesions other than follicular adenoma, the value for the area under the ROC curve (AUROC) was 87.7% (*p* < 0.001) (Figure 1A). The threshold probability for making diagnosis was then calculated to be 0.535 based on the optimum cut-off point on the ROC curve closest to (0,1). The corresponding diagnosis sensitivity, specificity, PPV, and NPV were 84.5%, 81.0%, 82.3%, and 82.5%, respectively. Figure 2 shows the differentiation performance of the logistic model using box plots for different cytological categories.

### 2.2. Performance of the Logistic Model in Cytologically Indeterminate Categories

The ROC curve analyses were performed for cases in the indeterminate AUS, FN/SFN, and SUSP cytological categories and their AUROC values were 81.3% (*p* < 0.001), 56.8% (*p* = 0.52), and 78.7% (*p* = 0.014), respectively (Table 4). The optimum threshold probabilities based on the cut-off point closest to (0,1) for the AUS, FN/SFN, and SUSP categories were calculated to be 0.523, 0.304, and 0.733, with (sensitivity percentage, specificity percentage) equal to (87.0, 70.6), (80.0, 45.5), and (64.0, 100.0), respectively (Table 4, model 2). The logistic model was more sensitive for the AUS category and was very specific for the SUSP category, while the model was not as effective for FN/SFN cases.

To assist the cytologist in further differentiating the benign cases from the indeterminate cases, one may choose the threshold probabilities to ensure a sensitivity of 100%, i.e., no malignant cases missed, with the specificity as high as possible. Table 4 shows the diagnosis results under various threshold probabilities set at different cut-off points of the ROC curves. With the cut-off points ensuring 100% sensitivity, it can be seen that the threshold probability values were 0.209, 0.075, and 0.426 for the AUS, FN/SFN, and SUSP categories and the specificities can reach 17.6%, 13.6%, and 33.3%, respectively (Figure 1 panel B, C, and D, Table 4 and model 1). With the assistance of the logistic model constructed by the computerized cytological features, 3 out of 17 benign AUS cases, 3 out of 22 benign FN/SFN cases, and 1 out of 3 benign SUSP cases might have avoided aggressive treatment, such as surgical thyroidectomy.

## 3. Discussion

This study proposed several new parameters for computerized morphometry of thyroid FNAC specimens. We can quantify not only morphological but also chromatic features. Our study shows that computerized quantification of cytological characteristics can help to differentiate 17.6%–33.3% of cases as benign rather than malignant, with 100% sensitivity, for the Bethesda categories of AUS and SUSP. Even for the FN/SFN category, 13.6% of cases can be further differentiated as benign with 100% sensitivity, using a model based on PTC cases and benign nodular/adenomatous goiter/hyperplasia. This is because the model is designed to differentiate not only PTC nodules but also benign nodular adenomatous goiter and hyperplasia. Among the three of 22 benign cases of FN/SFN differentiated with the model, one was an adenomatous goiter and two were hyperplastic nodules. The cytological features used in this model for differential diagnosis included mean nuclear size (MNS), mean nuclear elongation (MNElon), nuclear-to-cytoplasmic saturation ratio (NCSR), nuclear-to-cytoplasmic ratio (NCR), nuclear polarity (NP), and inclusion index (II).

Much effort has focused on genetic analysis to aid in diagnosis of indeterminate cytology reports for thyroid nodules. *BRAF*, *NRAS*, *HRAS*, and *KRAS* point mutations, and *RET/PTC1*, *RET/PTC3*, and *PAX8/PPARc* rearrangements were reported to be helpful [6]. Several commercial thyroid FNAC molecular tests are available, including ThyroSeq, Afirma, RosettaGX Reveal, ThyGenX, and ThyraMIR [9]. However, genetic examination is expensive and not widely available. We previously demonstrated that computerized morphometry of FNAC samples helps predict recurrence of PTC [7]. The nucleus-cell ratio and variation of nuclear area showed significant and positive correlations with recurrence (*p* = 0.002 and 0.044, respectively). The present study quantified cytological features and identified differences between benign and malignant thyroid nodules. Larger nuclear size and higher nuclear-to cytoplasmic ratio are related to malignancy. This may be related to cellular duplication activity [8], and was reportedly associated with prognosis for several cancers, including malignant melanoma, rhabdomyosarcoma, and renal cell carcinoma [15,16,17]. Our study also showed that NCSR is related to diagnosis. PTC cells show less difference in saturation between nuclei and cytoplasm than benign follicular cells. There has been no other description of the association between this chromatic feature and malignancy in the literature. PTC cells had greater mean nuclear elongation than benign follicular cells in our study. This also had not been mentioned in the research. PTC cells are usually found in a papillaroid arrangement. In the cross section of each papilla, the elongated cells are arranged in a spiral shape, which results in a different direction for each cell and greater nuclear polarity in PTC. Inclusion bodies are characteristic features of PTC, and, therefore, the index of inclusion is higher in PTC.

This computerized diagnostic score was developed for the purpose of assisting cytopathologists and clinicians in challenging situations of indeterminate thyroid cytology. According to the American Thyroid Association guideline, in the situation of indeterminate thyroid cytologic results, clinicians should discuss with the patients to decide further management, including repeated FNAC, genetic study, or thyroidectomy [1]. In our dataset of AUS cases, we can set the cut-off point of the diagnostic score at 0.523 to get the best overall diagnostic power (sensitivity: 87%, specificity: 70.6%) for the differentiation of benign and malignant tumors (Table 4). Less aggressive management such as repeated FNAC may be preferred for cases with diagnostic scores less than 0.523. In contrast, more aggressive management such as thyroidectomy may be preferred for patients with diagnostic scores more than 0.523. Or we can set the cut-off point at 0.209 to get the best sensitivity (100%) for the purpose of avoiding unnecessary operation, which, in our dataset, avoids 13.6% (3 out of 22) of benign FN/SFN cases in our dataset from aggressive treatment. This scenario may also be used in the situation with the cytological diagnosis result of suspicions of malignancy (SUSP). We can set the diagnostic score of computerized cytomorphometry at 0.733 to get the best overall diagnostic power (sensitivity: 64%, specificity: 100%) (Table 4). We could set the cut-off point at 0.426 to get the best sensitivity (100%) while avoiding extensive thyroidectomy, i.e., 33% (1 out of 3) of benign SUSP cases in our dataset, could have been exempted from aggressive treatment. In the situation of follicular neoplasm (FN)/suspicious of FN (SFN), the overall diagnostic power for differentiation of follicular adenoma (FA) and follicular thyroid carcinoma (FTC) is poor (Table 4). The comparison of the diagnostic score of FA and FTC was shown in Figure 2C. The ROC curve of the diagnostic score to differentiate benign and malignant lesions in FN/SFN was shown in Figure 1C with the AUC of 0.568 and *p* value of 0.52. However, if we set the cut-off point of the diagnostic score at 0.075 to get the best sensitivity (100%), we could still avoid 13.6% (3 out of 22) benign FN/SFN cases in our dataset from aggressive treatment.

The cytological features that can differentiate benign from malignant thyroid tumors in AUS/FLUS and SUSP categories of cytological specimens do not perform well for the category of FN/SFN. This indicates that the cytological characteristics of tumors with a follicular growth pattern are different from those of tumors with a papillary growth pattern. This is supported by the fact that the oncogenes associated with these two growth patterns are also different [18]. Since the diagnosis of follicular neoplasm depends on the histologic analysis of vascular and/or capsular invasion, the literature has reported that it is difficult to differentiate benign from malignant follicular neoplasms in cytological specimens [6] and it is not surprising to see that the computerized cytological features proposed by this study have also failed to differentiate the follicular neoplasms effectively.

In our study, the Red-Green-Blue (RGB) color space of image pixels were first converted into the color space of hue (H), saturation (S), and value (V). To the best of our knowledge, this is the first study to evaluate the thyroid FNAC with indeterminate categories using the color space of HSV. In the study conducted by Celik ZE et al., morphological and chromatic variables were used to establish their computerized cytomorphometry system, which included five variables [19]. Among these five variables, only one was chromatic variable and was in grey scale. In another study [20], Gilshtein H et al. applied 50 variables in their computerized cytomorphometry, but the texture of nuclear chromatin was also in gray scale. In addition, our study collected a large dataset of 240 cases. Prior research included 125 cases of indeterminate cytological categories while the previous studies had only 40 and 58 cases, respectively.

The major limitation of this study is the small number of patients in each Bethesda category. In our dataset, more than 95% of PTC (105 out of 110) were cPTC and only five cases were follicular variants of PTC. We have noted that the cPTC constitutes 80% of all PTC in the literature [21] and further study is needed especially for the non-invasive follicular thyroid neoplasm with papillary like nuclear features (NIFTP) because of its different malignant potential [22]. For a more comprehensive clinical use, we should also collect a sufficient sample of other types of thyroid cancers, such as anaplastic thyroid cancer, poorly differentiated thyroid cancer, and medullary thyroid cancers, to be included in the analysis. However, the findings of the current study are still of value in differential diagnosis of PTC, which constitutes most thyroid cancers. Another concern is the use of Riu’s stain, which is not widely used in other countries, and the proposed cytological features should be further evaluated using different stains. However, since the Riu’s stain, which is also a type of Romanowsky stain [23], has the appearance resembling the Diff-Quik stain [24]. Similar results may be anticipated for the use of the Diff-Quik stain. However, different staining methods may result in different NCHR, NCSR, and NCVR values. We should collect and analyze specimens of other staining methods such as the Papanicolaou stain in the future.

## 4. Materials and Methods

This study followed the ethical principles of the Declaration of Helsinki and was approved by the Institutional Review Board of the National Taiwan University Hospital (NTUH) (protocol number: 201603064RIPB). The study protocol with the waiver of informed consent has been approved by NTUH’s research ethics committee and was registered on ClinicalTrials.gov (protocol number: NCT03105648).

### 4.1. Study Subjects

We recruited 244 cases who had pathologic diagnoses of benign nodules, PTC, follicular thyroid carcinoma (FTC), or Hurthle cell carcinoma, and had underwent FNAC by endocrinologists in NTUH between 2010 and 2017, which was followed by thyroidectomy. Three cases with a pathologic diagnosis of lymphocytic thyroiditis and 1 with a pathologic diagnosis of Graves’ disease were excluded (Figure 3). The analysis of 240 cases included 110 cases with PTC, 100 with nodular/adenomatous goiter/hyperplasia, 10 with follicular/Hurthle cell carcinoma, and 20 with follicular adenoma.

FNAC specimens underwent Riu’s staining, which is a type of Romanowsky staining [23]. Riu’s staining was performed in the following standard procedure. After air drying, the smears were covered with solution A (0.05% methylene blue and 0.17% eosin Y in methanol) for 30 s, which was followed by placement of solution B (0.12% azure I, 0.14% methylene blue, 2.52% Na_2_HPO_4_·12H_2_O, and 1.25% KH_2_PO_4_ in distilled water) for 90 s. The stain mixtures were then washed off and the slides were observed under a light microscope. Digital images of these stained FNAC specimens were obtained and analyzed.

### 4.2. Computerized Analysis of Cytologic Features

Digital cytological images in matrices of color pixels were collected for computerized analysis (Figure 4).

The image processing and analysis were performed using AmCAD-CA (AmCad BioMed Corp., Taipei, Taiwan). The digital cytological images were segmented into three clusters, including background, cytoplasm, and nucleus, based on the pixel value in hue and saturation with Otsu’s method. After segmentation, the background cluster was set to zero, and the Canny edge detection was performed on the remaining image to find the boundary of nuclear area. The detailed algorithm used by the software can be seen in the patent [25] filed by the software company. The pixel values in the Red-Green-Blue (RGB) color space were first converted into the color space of hue (H), saturation (S), and value (V). The H of nuclei and cytoplasm in the same cytologic specimen was compared and the difference was calculated in a nuclear-to-cytoplasmic hue ratio (NCHR) using a formula presented in the Appendix A. Nuclear-cytoplasmic saturation ratio (NCSR) and nuclear-cytoplasmic value ratio (NCVR) were generated in similar fashion. Nuclear-cytoplasmic ratio (NCR) was defined as the ratio of the area of the nuclei and cytoplasm. The segmented margin of the nuclei could then be used for statistical values, such as the sample mean (M), the sample standard deviation (SD), and the coefficient of variation (CV = SD/M) of the morphological features including nuclear size, circularity, ellipticity, elongation, nuclear polarity, inclusion, and overlapping. Circularity indicated roundness of a cell nucleus by considering the nuclear perimeter and area. Ellipticity evaluated nuclear surface smoothness by considering axis lengths, perimeter, and area of the nucleus. Elongation assessed nuclear extension by considering both major and minor axis lengths of the nucleus. The overlapping index was defined as the ratio of overlapped nuclei to the total nuclear area. The index of nuclear polarity (NP) was defined as the variability of nuclear long-axis angles. The above calculating formulae were listed in the Appendix A.

### 4.3. Statistical Analysis

A total of 22 quantified cytological features, including 16 morphological and 6 chromatic features, were computed and analyzed. Analysis of variance and the Scheffé test were used to compare the differences in cytological features among 4 types of thyroid nodules based on a pathologic diagnosis. The D’Agostino-Pearson test was used to test for normal distribution of each feature. Multivariate logistic regression analysis was used to investigate the relationship between cytological features and pathologic results. Statistical analysis was performed using MedCalc Version 17.6 (MedCalc Software, Mariakerke, Belgium) and Stata/SE 14.0 for Windows (StataCorp LP, TX, USA). A *p* value less than 0.05 was considered statistically significant.

## 5. Conclusions

Our study showed that quantification of cytological morphological and chromatic features could be used to develop a computer-aided tool for diagnosing thyroid nodules. We will recruit more cases and other types of thyroid cancer to improve the diagnostic scoring system and examine the cytological features of specimens prepared with other staining methods.

## Figures and Tables

**Figure 1 cancers-11-01645-f001:**
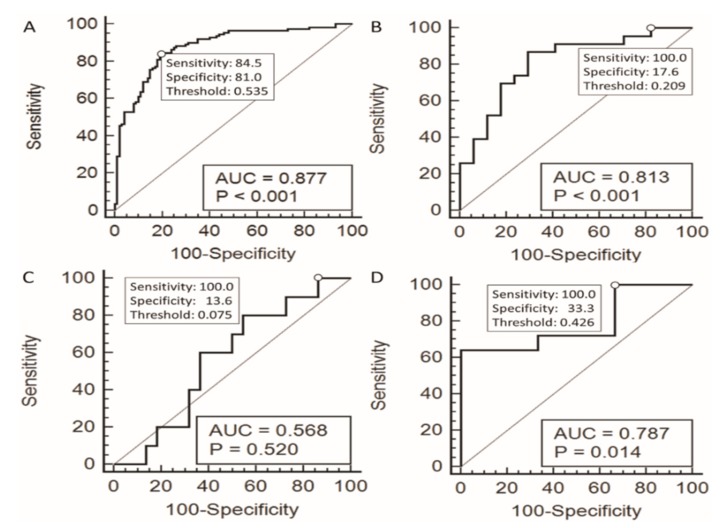
Receiver operating characteristic curves and their area under the curve (AUC), where positive and negative outcomes were defined as malignant and benign, respectively. (**A**) In pathologic diagnosis of papillary thyroid cancer and benign follicular lesions other than follicular adenoma (*n* = 210). (**B**) In Bethesda cytological category of atypia of undetermined significance or follicular lesion of undetermined significance (*n* = 40). (**C**) In Bethesda cytological category of follicular neoplasm or suspicious for a follicular neoplasm (*n* = 32). (**D**) In Bethesda cytological category of suspicious for malignancy (*n* = 53). In panel A, the threshold value was set at the cut-off point that was closest to (0,1). In panel B, C, and D, the threshold value was set at the sensitivity that was 100% aligned with specificity that is as high as possible.

**Figure 2 cancers-11-01645-f002:**
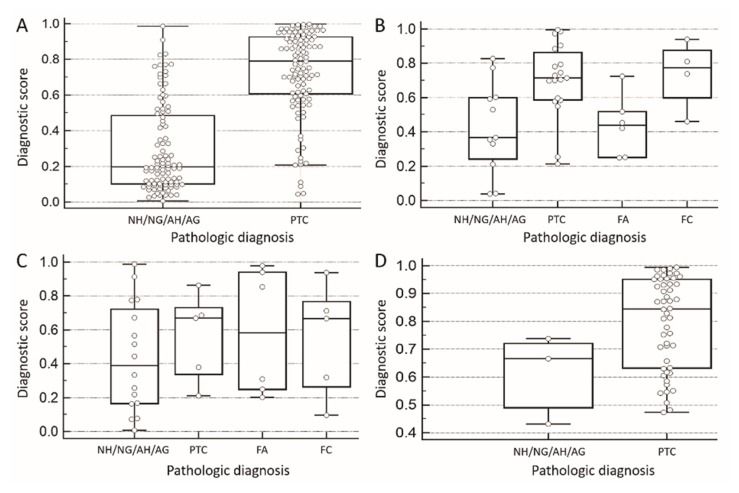
Box plots of the relationship between a diagnostic score and pathologic diagnosis. (**A**) In pathologic diagnosis of papillary thyroid cancer and benign follicular lesions other than follicular adenoma (*n* = 210). (**B**) In Bethesda cytological category of atypia of undetermined significance or follicular lesion of undetermined significance (*n* = 40). (**C**) In Bethesda cytological category of follicular neoplasm or suspicious for a follicular neoplasm (*n* = 32). (**D**) In Bethesda cytological category of suspicions for malignancy (*n* = 53). NH: Nodular hyperplasia. NG: Nodular goiter. AH: Adenomatous hyperplasia. AG: Adenomatous goiter. PTC: Papillary thyroid cancer. FA: Follicular adenoma. FC: Follicular carcinoma.

**Figure 3 cancers-11-01645-f003:**
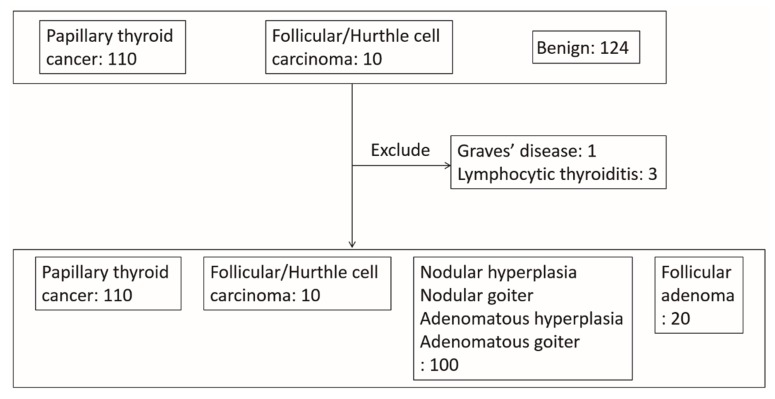
Flowchart of subject recruitment.

**Figure 4 cancers-11-01645-f004:**
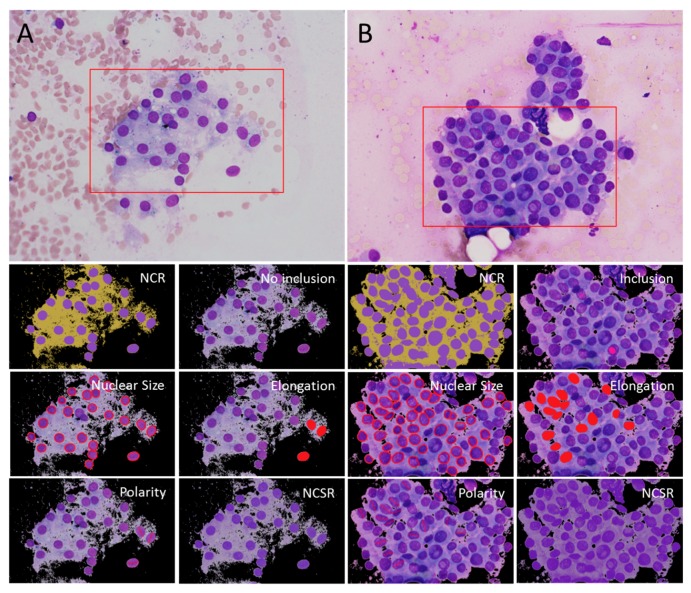
Visual comparison for clinically important features. The images from a case of hyperplasia (**A**) and a case of papillary thyroid carcinoma (**B**) acquired from a thyroid fine-needle aspiration cytology smear (Riu’s stain, 400×). The image (red rectangle region of interest) was cropped for computerized analysis. NCR (nuclear-cytoplasmic ratio visualized with nuclei in purple, cytoplasm in yellow), intranuclear inclusion (pink), nuclear size (nuclei segmented by red contours), elongation (elongated nuclei in red), polarity (alignment of red major axes of elongated nuclei), and NCSR (nuclear-cytoplasmic saturation ratio visualized with a saturation contrast between nuclei and cytoplasm).

**Table 1 cancers-11-01645-t001:** Basic characteristics of the study subjects.

Cytological Diagnosis (Bethesda Category)	Benign			Atypia of Undetermined Significance or Follicular Lesion of Undetermined Significance (AUS)			Follicular Neoplasm or Suspicious for a Follicular Neoplasm (FNSFN)			Suspicious for Malignancy (SUSP)			Malignant
Pathologic Diagnosis	Benign	Malignant (%)	*p*	Benign	Malignant (%)	*p*	Benign	Malignant (%)	*p*	Benign	Malignant (%)	*p*	Malignant (%)
Number	78	12 (13.33%)		17	23 (57.5%)		22	10 (31.25%)		3	50 (94.34%)		25 (100%)
Age (SD)	52 (12)	50 (19)	0.525	46 (14)	44 (11)	0.564	55 (12)	54 (20)	0.817	50 (29)	52 (11)	0.868	51 (15)
Gender (Female: Male)	69:9	10:2	0.629	13:4	16:7	0.645	18:4	6:4	0.215	3:0	38:12	0.355	22:3

*p*-values were calculated by independent samples *t*-test for the age and by Mann-Whitney test for the gender.

**Table 2 cancers-11-01645-t002:** Cytomorphometry of fine needle aspiration specimens in different pathologic diagnosis.

Pathologic Diagnosis	Papillary Thyroid Cancer and Benign Follicular Lesions other than Follicular Adenoma		Follicular Neoplasm		*p* *
	Nodular Hyperplasia/Goiter, Adenomatous Hyperplasia/Goiter (Group 1)	Papillary Thyroid Cancer (Group 2)	Follicular Adenoma (Group 3)	Follicular Carcinoma Hurthle Cell Carcinoma (Group 4)	
Number	100	110	20	10	
Age (year)	53 (13)	50 (13)	48 (14)	48 (19)	0.358
Gender (Female: Male)	87:13	85:25	16:4	7:3	0.247
Mean nuclear size (μm^2^)	56.03 (43.46–74.93)	75.07 (65.67–87.15)	81.42 (60.69–95.13)	77.60 (58.38–98.14)	<0.001 ^†,‡,§^
Standard deviation of nuclear size (μm^2^)	8.02 (5.06–11.23)	10.57 (8.16–14.36)	12.92 (5.90–16.02)	12.71 (8.95–20.16)	0.007 ^†^
Coefficient of variation of nuclear size	0.150 (0.095–0.214)	0.144 (0.114–0.192)	0.161 (0.077–0.292)	0.163 (0.137–0.180)	0.927
Mean nuclear circularity	0.839 (0.819–0.851)	0.832 (0.817–0.840)	0.846 (0.831–0.861)	0.839 (0.811–0.852)	0.062
Standard deviation of nuclear circularity	0.019 (0.014–0.029)	0.021 (0.015–0.031)	0.014 (0.011–0.023)	0.020 (0.015–0.024)	0.632
Coefficient of variation of nuclear circularity	0.023 (0.016–0.036)	0.025 (0.018–0.038)	0.016 (0.013–0.027)	0.024 (0.018–0.030)	0.672
Mean nuclear elongation	0.505 (0.457–0.559)	0.546 (0.503–0.569)	0.478 (0.453–0.541)	0.506 (0.486–0.572)	<0.001 ^†,||^
Standard deviation of nuclear elongation	0.067 (0.049–0.080)	0.062 (0.051–0.077)	0.066 (0.046–0.098)	0.069 (0.054–0.078)	0.11
Coefficient of variation of nuclear elongation	0.134 (0.099–0.165)	0.115 (0.092–0.141)	0.133 (0.098–0.196)	0.120 (0.108–0.154)	0.012 ^†^
Mean nuclear ellipticity	0.872 (0.852–0.881)	0.871 (0.860–0.879)	0.880 (0.864–0.892)	0.876 (0.861–0.889)	0.017 ^‡^
Standard deviation of nuclear ellipticity	0.014 (0.010–0.021)	0.015 (0.011–0.020)	0.014 (0.008–0.019)	0.014 (0.012–0.023)	0.528
Coefficient of variation of nuclear ellipticity	0.015 (0.011–0.024)	0.017 (0.012–0.024)	0.015 (0.009–0.022)	0.016 (0.014–0.027)	0.556
Nuclear-to-cytoplasmic hue ratio	1.024 (1.011–1.043)	1.036 (1.016–1.053)	1.015 (0.998–1.026)	1.027 (1.012–1.053)	0.001 ^||^
Standard deviation of Nuclear-to-cytoplasmic hue ratio	0.985 (0.864–1.087)	0.983 (0.885–1.116)	0.901 (0.822–0.984)	0.902 (0.830–0.990)	0.082
Nuclear-to-cytoplasmic saturation ratio	1.944 (1.652–2.422)	1.693 (1.570–1.979)	2.232 (1.914–2.347)	2.322 (1.731–2.566)	<0.001 ^†,||^
Standard deviation of Nuclear-to-cytoplasmic saturation ratio	0.911 (0.797–1.109)	0.908 (0.774–1.013)	0.937 (0.813–1.152)	0.920 (0.848–0.991)	0.105
Nuclear-to-cytoplasmic value ratio	0.833 (0.791–0.872)	0.835 (0.805–0.858)	0.798 (0.756–0.863)	0.864 (0.789–0.869)	0.232
Standard deviation of Nuclear-to-cytoplasmic value ratio	1.086 (0.935–1.194)	1.099 (0.986–1.286)	0.987 (0.933–1.158)	1.133 (0.945–1.227)	0.506
Nuclear-to-cytoplasmic ratio	0.998 (0.740–1.373)	1.230 (1.020–1.556)	1.056 (0.755–1.806)	0.941 (0.806–1.114)	0.025 ^†^
Nuclear polarity	14.02 (5.04–23.49)	22.48 (15.27–32.69)	13.09 (4.19–18.26)	12.75 (7.56–19.01)	<0.001 ^†,||^
Inclusion index	0 (0.000–0.004)	0.006 (0.002–0.012)	0.004 (0.000–0.007)	0.007 (0.005–0.018)	0.011 ^†^
Overlapping index	0.436 (0.340–0.577)	0.500 (0.432–0.589)	0.473 (0.290–0.581)	0.452 (0.374–0.596)	0.221

Data are presented as mean ± SD if the continuous variable is normally distributed, and as a median (interquartile range) if not normally distributed. * P was obtained using the ANOVA test to compare the difference between the four groups. ^†^ The results are significantly different between group 1 and group 2, ^‡^ significantly different between group 1 and group 3, ^§^ significantly different between group 1 and group 4, and ^||^ significantly different between group 2 and group 3 when using the Scheffé test for all pairwise comparisons.

**Table 3 cancers-11-01645-t003:** The relationship between pathologic diagnosis and cytological characteristics in pathologic diagnosis of papillary thyroid cancer and benign follicular lesions other than follicular adenoma. Multivariate logistic regression analysis was performed using pathological diagnosis as the dependent variate (malignant: 1, benign: 0) and six cytological characteristics as independent variates (*n* = 210).

Variable	Coefficient	*p*
Mean nuclear size (μm^2^)	0.06557	<0.0001
Mean nuclear elongation	8.94005	0.0239
Nuclear-to-cytoplasmic saturation ratio	−1.27953	0.0008
Nuclear-to-cytoplasmic ratio	1.89500	<0.0001
Nuclear polarity	0.038098	0.0259
Inclusion index	32.19870	0.0473
Constant	−9.71158	0.0001

**Table 4 cancers-11-01645-t004:** Prediction of pathologic diagnosis of thyroid cancer using a diagnostic score.

Cytological Diagnosis	AUROC (*p*-Value)	Model 1: Threshold Value was Set at the Sensitivity that was 100% with the Specificity as High as Possible					Model 2: Threshold Value was Set at the Cut-off Point on the ROC Curve that was Closest to (0,1)				
		Sensitivity	Specificity	PPV	NPV	Threshold Value	Sensitivity	Specificity	PPV	NPV	Threshold Value
AUS	0.813 (<0.001)	100.00%	17.60%	62.20%	100.00%	0.209	87.00%	70.60%	80.00%	80.00%	0.523
FN/SFN	0.568 (0.520)	100.00%	13.60%	34.50%	100.00%	0.075	80.00%	45.50%	40.00%	83.30%	0.304
SUSP	0.787 (0.014)	100.00%	33.30%	96.20%	100.00%	0.426	64.00%	100.00%	100.00%	14.30%	0.733

PPV: positive predictive value. NPV: negative predictive value. AUS: atypia of undetermined significance or follicular lesion of undetermined significance. FN/SFN: follicular neoplasm or suspicious for a follicular neoplasm. SUSP: suspicious for malignancy.

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
