# Peer review of "Computerized Cytological Features for Papillary Thyroid Cancer Diagnosis—Preliminary Report"

_cancers, 2019, doi:10.3390/cancers11111645_

Round 1
Reviewer 1 Report
The paper is interesting however major clinical weaknesses are easily found, probably due to some problems in the material and methods section, that should be clarified.
The model is designed to differentiate not only PTC nodules but also benign nodular adenomatous goiter and hyperplasia. Moreover other histotypes are not included: MTC, rare neoplasms. In particular what about NIFTP subgroup?
Please show to the readers a direct comparison between particularly challenging situations like follicular adenoma Vs FTC ( does the model work?). or hyperplastic nodules Vs NIFTP.
PTC is a dramatically heterogeneous diagnostic category: how many classic? How many follicular variant did they include?
Please, avoid pathogenetic suppositions as “ Whether the increased cytoplasmic saturation is due to an increase in organelles or cellular products of protein and thus correlates with malignancy needs further research. PTC cells had greater mean nuclear elongation than benign follicular cells in our study. This is probably because mitosis would result in elongated nuclear shape and PTC cells show higher rates of mitosis.”.
It is very dangerous in our experience to put attention to mitoses in thyroid pathology, except from the Turin criteria for undifferentiated thyroid carcinoma.
Figure 4 is quite intriguing, however the reader should be helped in understanding better the various computational morphological criteria (elongation, Nuclear-to-cytoplasmic saturation ratio, polarity)
The paper is direct to clinicians, so the mathematical model should be included as supplementary data, since it is very difficult to understand.
On the contrary, the discussion should include the limitations or the possible “plus” in applying the model to really controversial entities like AUS-FLUS and SFN.
Author Response
Dear Reviewer,
Thank you very much for offering us a chance to revise our manuscript and giving us your constructive comments. We have revised the manuscript according to the suggestions. All of the questions have been carefully addressed in the attachment, and all changes are highlighted with yellow color background in the revised manuscript.
Thank you again for your precious comments!
Sincerely yours,
Argon Chen, Ph.D.
Institute of Industrial Engineering
National Taiwan University
E-mail: [email protected]
Tel: 886-2-3366-9504
Fax: 886-2-2362-5856

Reviewer 2 Report
This study aimed to analyze the cytological features of benign and malignant thyroid nodules with the use of computerized cytomorphometry for the differential diagnosis of FNAC. Similar studies have been reported (Celik ZE, et al. Using Computerized Cytomorphometry to Distinguish between Benign and Malignant Cases in Thyroid Fine-Needle Aspiration Cytology. Diagn Cytopathol. 2016 Nov;44(11):902-911. doi: 10.1002/dc.23611) (Gilshtein H, et al. Computerized cytometry and wavelet analysis of follicular lesions for detecting malignancy: A pilot study in thyroid cytology. Surgery. 2017 Jan;161(1):212-219. doi: 10.1016/j.surg.2016.06.078). The authors should clarify the differences between their study and the previous studies.
Lines 212-216, would the color value of the pixels of the image be varied due to potential variations in the staining techniques? Please clarify the consistency of the staining techniques and how these would affect the image analysis.
It is unclear that which software was used for the image analysis. MATLAB? More information should be provided for the segmentation of nuclei. How the Canny edge detection method incorporated in the image processing software? Is any deep learning method used in the segmentation? To increase the transparency of the method, more information about the image analysis should be included.
According to the footnote of Table 2, statistical tests were performed to evaluate whether the data were normally distributed or not. However, the statistical tests used were not described in the methodology. Please include this information in the section “Statistical Analysis”.
This study did not compare the diagnostic performance between computer-based method with the pathologist-based method in the assessment of thyroid FNAC specimens. In the study, the result showed that the computerized quantification method yielded 100% sensitivity and 17.6-33.3% specificity. However, if the pathologist-based method achieves better result than the computerized quantification method, the clinical application of the computerized method is questionable. The authors should consider comparing the result of pathologist assessment with the computerized method, and discuss the additive value of using computerized method.
The authors should discuss the value of using computerized method in routine clinical practice. Should the computerized method be used for initial screening of the specimens before the pathologist’s assessment, or it is used after the pathologist’s assessment?
Author Response

(The authors gave the same response as above.)

Round 2
Reviewer 2 Report
The responses to the questions and the revision of the manuscript are satisfactory. I have no further comments.